# Prevalence and determinants of unintended pregnancy in Ethiopia: A systematic review and meta-analysis of observational studies

**Muluneh Alene[ID]¹\*, Leltework Yismaw¹, Yebelay Berelie², Bekalu Kassie³, Reta Yeshambel⁴, Moges Agazhe Assemie[ID]¹**

1 Department of Public Health, Debre Markos University, Debre Markos, Ethiopia, 2 Department of Statistics, Debre Markos University, Debre Markos, Ethiopia, 3 Department of Midwifery, Debre Markos University, Debre Markos, Ethiopia, 4 Department of Biology, Mizan-Tepi University, Teppi, Ethiopia

\* mulunehadis@gmail.com

## Abstract

### Background

Unintended pregnancy has significant consequences for the health and welfare of women and children. Despite this, a number of studies with inconsistent findings were conducted to reduce unintended pregnancy in Ethiopia; unavailability of a nationwide study that determines the prevalence of unintended pregnancy and its determinants is an important research gap. Thus, this study was conducted to determine the overall prevalence of unintended pregnancy and its determinants in Ethiopia.

### Methods

We searched from Google Scholar, PubMed, Science Direct, Web of Science, CINAHL, and Cochrane Library databases for studies. Each of the original studies was assessed using a tool for the risk of bias of observational studies. The heterogeneity of studies was also assessed using $I^2$ test statistics. Data were pooled and a random effect meta-analysis model was fitted to provide the overall prevalence of unintended pregnancy and its determinants in Ethiopia. In addition, the subgroup analyses were performed to investigate how the prevalence of unintended pregnancy varies across different groups of studies.

### Results

Twenty-eight studies that satisfy the eligibility criteria were included. We found that the overall prevalence of unintended pregnancy in Ethiopia was 28% (95% CI: 26–31). The subgroup analyses showed that the highest prevalence of unintended pregnancy was observed from the Oromiya region (33.8%) followed by Southern Nations Nationalities and Peoples' region (30.6%) and the lowest was in Harar. In addition, the pooled prevalence of unintended pregnancy was 26.4% (20.8–32.4) and 30.0% (26.6–33.6) for community-based cross-sectional and institution-based cross-sectional studies respectively. The pooled analysis showed that not communicating with one's husband about family planning was more likely to lead to unintended pregnancy (OR: 3.56, 95%CI: 1.68–7.53). The pooled odds ratio

**Data Availability Statement:** All relevant data are within the paper and its Supporting Information files.

**Funding:** The author(s) received no specific funding for this work.

**Competing interests:** The authors have declared that no competing interests exist.

also showed that unintended pregnancy is more likely among women who never use family planning methods (OR: 2.08, 95%CI: 1.18–3.69). Furthermore, the narrative review of this study showed that maternal education, age, and household wealth index are strongly associated with an unintended pregnancy.

## Conclusions

In this study, the prevalence of unintended pregnancy was high. Lack of spousal communication, never using family planning, maternal education, and household wealth level were significantly associated with an unintended pregnancy. This study implies the need to develop plans and policies to improve the awareness of contraceptive utilization and strengthen spousal communication related to pregnancy.

## Introduction

Unintended pregnancy is a pregnancy which is either mistimed or unwanted [1,2]. It is a public health problem and a risk factor for adverse health outcomes, particularly for maternal and child health [3]. Though the rate of unintended pregnancy fell worldwide between 1990 and 2014, it dropped less sharply in developing regions than in developed regions [3]. Ethiopia is one of the developing countries with a high prevalence of unintended pregnancy. In Ethiopia, more-than one-third (38%) of pregnancies were unintended in 2014; slightly lower than in 2008 which was 42%. According to results from the 2016 Ethiopian Demographic and Health Survey (EDHS) of all births in the past five years and current pregnancies, 25% are unintended. Also, the 2016 EDHS report showed that the overall difference between the wanted fertility rate and the total fertility rate is one child, which suggests that Ethiopian women are currently having, on average, one child more than they want [4].

Pregnancies should be planned before conception; otherwise, a woman may not be in optimal health for childbearing [2]. Unintended pregnancy leads to maternal mortality and morbidity due to the complications of unsafe abortion, miscarriage, and unplanned births, which burdens the health system at all [5–7]. Annually, more than 1 in 10 pregnancies end in abortion, and 1 in 27 mothers die due to the complications of pregnancy or childbirth in Ethiopia [8]. A woman with an unintended pregnancy is more likely to have low physical and mental health, low self-care, and depression during pregnancy. These lead to poor Antenatal Care (ANC) service utilization and postpartum depression, which is risky for unfavorable pregnancy outcome, and maternal morbidity and mortality [7,9]. Consequently, the newborns of unintended pregnancies are faced with low birth weight and inadequate vaccinations which increases the risk for childhood illnesses [5,7,10–12]. Previous evidence showed that unintended pregnancy mainly results from inconsistent or incorrect use of contraceptive methods, and women are less likely than men to want more children no matter how many children they already have [2,13].

In Ethiopia, a number of studies were conducted to estimate the magnitude and to identify the determinants of unintended pregnancy. However, the reported prevalence and determinants in these fragmented studies vary depending on the characteristics of study participants, the type of design employed and the variables analyzed. Combined findings of existing studies significantly strengthen the quality of evidence investigating the national prevalence and determinants of unintended pregnancy. Thus, this systematic review and meta-analysis was conducted to determine the overall prevalence and determinants of unintended pregnancy in

Ethiopia. This review is intended to bring an improvement in the design of future studies related to unintended pregnancy. The findings of this study are also intended to improve health workers' interventions in the area of reproductive health.

## Materials and methods

### Study design and setting

A systematic review and meta-analysis, which aimed to estimate the overall prevalence of unintended pregnancy and its determinants was conducted in Ethiopia. Ethiopia is situated in the horn of Africa, and bordered by Eritrea to the north, Sudan and South Sudan to the west, Kenya to the south, and Djibouti and Somalia to the east. Nearly eight in ten women (78%) live in rural areas in Ethiopia, and half of women age 15–49 (48%) have no education [4].

### Eligibility criteria

In this systematic review and meta-analysis, studies were included with the following criteria: 1) only studies conducted in Ethiopia. 2) Only studies reported in English language. 3) Only studies involving pregnant women or women had given birth at least once preceding the survey. 4) All observational studies reporting the prevalence and determinants of unintended pregnancy. Both published and unpublished articles were included. Studies, which were not fully accessible after at least two-email contact with the primary authors, were excluded, because of the inability to assess the quality of studies without their full text.

### Searching for studies

A comprehensive search strategy was done by three (MA, LY, and RY) of the authors. Both published and unpublished articles on unintended pregnancy were searched from international (Google Scholar, PubMed, Science Direct, Web of Science, CINAHL, and Cochrane Library), and national (Ethiopian Journal of Public Health and Nutrition) electronic databases. First, articles were searched by examining the full titles ("Prevalence and determinants of unintended pregnancy in Ethiopia") and then keywords (unintended pregnancy, unplanned pregnancy, unwanted pregnancy, mistimed pregnancy, determinants, risk factors, associated factors, Ethiopia). These keywords were used separately and in combination using Boolean operators "OR" or "AND". In addition, we searched from the reference lists of all the included studies (snowball technique) to identify any other studies that may have been missed by our search strategy. Finally, all studies were imported into reference management software (Mendeley Desktop).

### Outcome measures and data extraction

Unintended pregnancy, which is either mistimed or unwanted, was the primary outcome of the study. Unwanted pregnancy occurred when a woman did not want to have any more pregnancies, whereas mistimed pregnancy is a pregnancy that was wanted by the woman at some time, but which occurred sooner than they wanted. The pregnancy intention for the included studies was assessed by interviewing current pregnant women or women who had given birth at least once preceding the survey.

All essential data from the included studies were extracted independently by two (MA and LY) of the authors using a predesigned data abstraction form. This form includes the last name of the first author, publication year, data collection period, study design, region of the study conducted, study population, sample size, response rate, and the magnitude of unintended pregnancy. In addition, we extracted the adjusted odds ratios with corresponding 95%

confidence intervals to measure the strength of effects. Deviation in a data extraction process was resolved by discussion and consensus involving all authors.

## Quality assessment for studies

The quality of meta-analysis depends on the included studies [14]. Two authors (MA and YB) assessed the risk of bias for the included studies using the tool of risk of bias assessment for observational studies [15]. This tool includes 10 items. The first four items assess the external validity, while the other six items evaluate the internal validity of the study. All items of the tool were filled in for each included study and categorized as low risk bias (if the response is "yes"), higher risk (if the response is "no"), and not clear. The quality of the study was determined by summing the score given for each item. Lastly, unclear risk of bias was categorized as high risk of bias then; the summary assessment risk of bias for each study was categorized according to the number of high risk of bias: low ($\geq$2), moderate (3–4), and high ($\geq$5).

## Data processing and analysis

After extracting all relevant data using Microsoft excel software, data were exported to R statistical software for meta-analysis. The double arcsine transformation which stabilizes the sampling variance was applied to estimate the weighted average prevalence, and the transformed summary prevalence are converted back for ease of interpretation [16].

We assessed the consistency of studies using $I^2$ test statistics [17]. This test examines the null hypothesis that all the included studies are evaluating the same effect. Consequently, since there was heterogeneity between the original studies ($I^2$ = 96%, p<0.01) a random effect model is needed, and to account for between-study variance a random effect meta-analysis with an estimation of DerSimonian and Laird method was performed. The possible sources of heterogeneity among studies might be differences in study participants, study design, risk of bias and data collection period. Furthermore, subgroup analyses were conducted to investigate how the prevalence of unintended pregnancy varied across different subgroups of studies.

## Results

### Search results

Our comprehensive search for studies was between the 15th of February and the 30th of April, 2019. The flow chart diagram that shows our literature search, study selection, and the number of included studies is presented in (**Fig 1**). Initially, a total of 273 articles were identified during our search, and then 158 articles were excluded due to duplication. Finally, 28 studies that satisfied the eligibility criteria were included in this systematic review and meta-analysis.

### Description of the included studies

The detail descriptions of the included studies are shown in (**Table 1**). All studies included in this review were published between 2006 and 2019. Of all the included studies, twelve [10,18,19–26,27,28] were community-based cross-sectional, while twelve were institutional-based cross-sectional studies [29,30,31–38,39]. The data for the two studies were also taken from the 2011 EDHS [40,41]. The majority of respondents of the original studies were pregnant mothers. The number of participants in each study varied from the lowest of 165 [29] to the highest of 7,759 [41]. A study with the smallest study participants was an institutional-based cross-sectional study, while the data for the highest study participants were taken from the 2011 EDHS. Furthermore, nine studies were conducted in the Amhara region, six studies

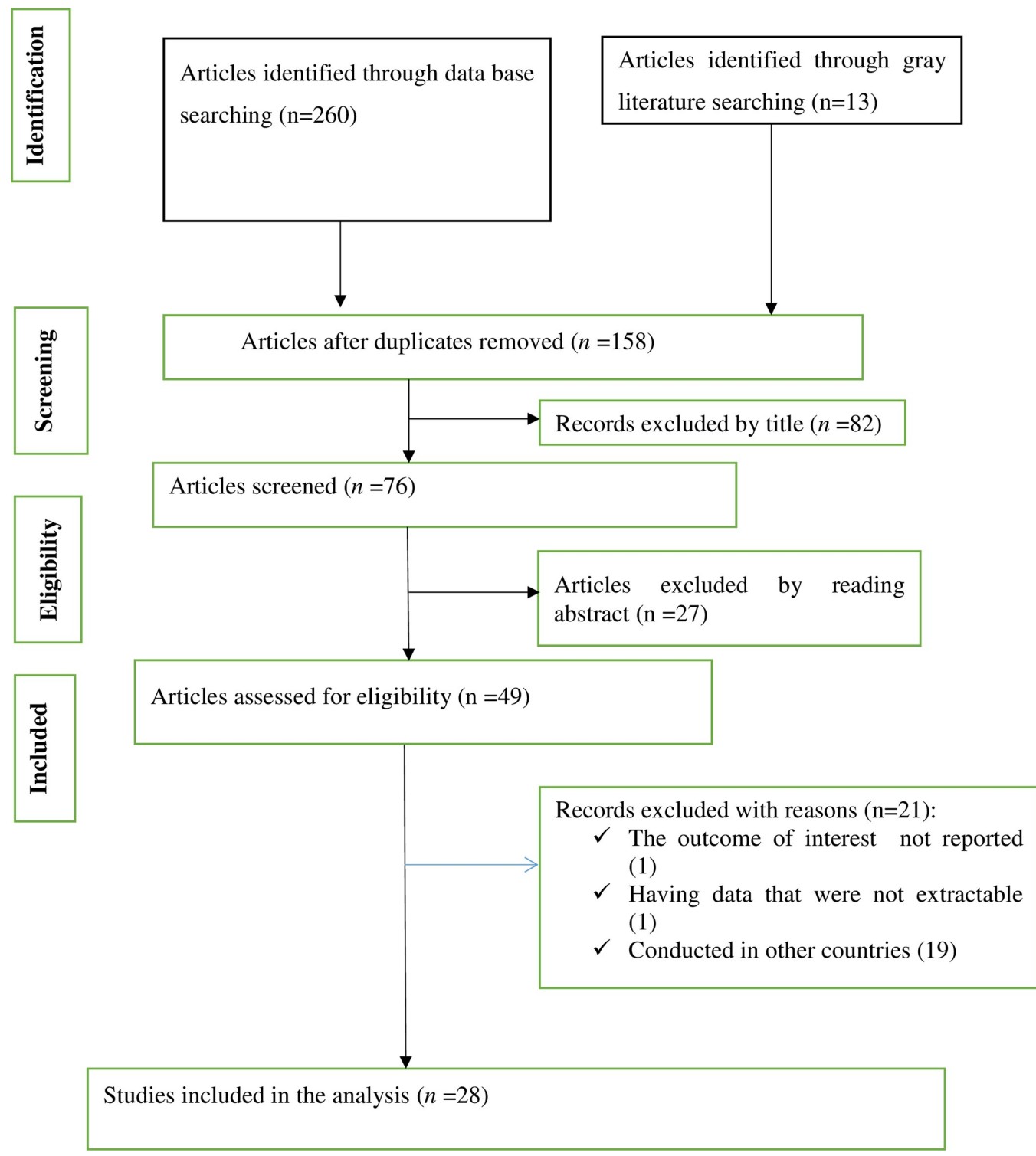

**Fig 1. Flow chart diagram describing the selection of studies included in the systematic review and meta-analysis of prevalence and determinants of unintended pregnancy in Ethiopia.**

**Table 1. Descriptions of the included studies conducted in Ethiopia on unintended pregnancy.**

| First author (publication year) | Study area (Region) | Study design | Data collection period | Study population | Sample size | Response rate | Prevalence of unintended pregnancy |
|---|---|---|---|---|---|---|---|
| Abame et al (2019) | SNNPR | CBCS | March 13, 2017 to April 13, 2017 | pregnant mothers | 748 | 97% | 36.2% |
| Abayu et al (2015) | Tigray | CBCS | 24/09/2012 to 18/10/2012 | Pregnant women | 626 | 96.3% | 26% |
| Admasu et al (2018) | Amhara | CBCS | NR | pregnant women or with under 1 child age | 680 | 91% | 15.8% |
| Ayele et al (2017) | Oromiya | IBCS | June 10, 2017 to July 24, 2017 | pregnant women following ANC | 165 | 96% | 35.2% |
| Darega et al (2015) | Oromiya | IBCS | May, 2014 | Women following ANC | 362 | 100% | 37.3% |
| Feyisso et al (2017) | SNNPR | IBCS | February to June, 2016 | Women following ante natal and post natal ANC | 290 | NR | 36.9% |
| Fite et al (2018) | Oromiya | CBCS | May 01–July 30, 2017 | Pregnant women | 704 | 91.5% | 41.5% |
| Gebreamlak et al (2014) | Amhara | IBCS | June to July 2012 | Pregnant women | 454 | NR | 26% |
| Gite et al (2016) | SNNPR | CBCS | February 15- March 11,2015 | Pregnant women | 311 | 95.4% | 19.4% |
| Gizaw et al (2018) | SNNPR | IBCS | February 24 to April 24, 2017 | Pregnant women | 224 | 100% | 22.3% |
| Goshu et al (2019) | Amhara | IBCS | April 01 to May 30, 2018 | Pregnant women | 398 | 100% | 26.1% |
| Habte et al (2013) | National level | EDHS | NA | Pregnant women | 1267 | NA | 24% |
| Hamdela et al (2012) | SNNPR | CBCS | April 02 to 15, 2011 | pregnant married women | 385 | 100% | 34% |
| Kahasay et al (2015) | Addis Abeba | IBCS | NR | female students aged from 16–19 years | 576 | 100% | 20.4% |
| Kassa et al (2012) | Oromiya | KDSHRC | December 2009 to November 2010 | Pregnant women | 2072 | 100% | 27.9% |
| Kassie et al (2017) | Addis Abeba | IBCS | February to May 2015 | Pregnant women | 393 | 100% | 36.4% |
| Kibret et al (2014) | Amhara | IBCS | April 15 to May 14, 2012 | Pregnant women | 413 | 100% | 32.9% |
| Liyew et al (2017) | Amhara | IBCS | NR | Pregnant women | 285 | 100% | 28.4% |
| Melese et al (2016) | Amhara | CBCS | NR | Pregnant women | 690 | NR | 23.5% |
| Mohammed et al (2016) | Oromiya | IBCS | January 10 to April 13, 2015 | Pregnant women | 413 | 97.9% | 27.1% |
| Mulat et al (2017) | SNNPR | IBCS | NR | Pregnant women | 362 | 100% | 33.7% |
| Tebekaw et al (2014) | National level | EDHS | EDHS, 2011 | Women who had at least one birth | 7,759 | NA | 32% |
| Teshome et al (2010) | Amhara | CBCS | NR | currently married women | 576 | NR | 40.8% |
| Tsegaye et al (2018) | Amhara | CBCS | August to September 2015 | Married pregnant women | 619 | 95.6% | 13.7% |
| Wado et al (2013) | Oromiya | HDSS | March,2012 | mothers with alive birth in the two years | 1456 | 94% | 35% |
| Worku et al (2006) | Harar | CBCS | November to December 2001 | Reproductive age women | 983 | 98.3% | 33.3% |
| Yenealem et al (2019) | Amhara | CBCS | April 1–May 30, 2014 | Pregnant women | 325 | 100% | 20.6% |

CBCS = Community-Based Cross-Sectional Study, EDHS = Ethiopia Demographic and Health Survey, HDSS = Health and Demographic Surveillance System, IBCS = Institution-Based Cross Sectional Study, KDS-HRC = Kersa Demographic Surveillance and Health Research Center, NA = Not Applicable, NR = Not Report, SNNRP = Southern Nations, Nationalities, and Peoples' Region

in the Oromiya region, six studies in Southern Nations, Nationalities, and Peoples' Region (SNNPR), two studies in Addis Ababa, two studies in Tigray, one study in Harar, and two studies from the national survey. Moreover, the prevalence of unintended pregnancy ranged from 13.7% [25] to 41.5% [21].

### Risk of bias assessment for the included studies

We assessed the risk of bias for each of the original studies using the existing risk-of-bias assessment tool (S1 Table). Of the total included studies, our summary assessment showed that more-than half (53.6%) of the studies had a low risk of bias, and one-fourth (25%) of studies had a moderate risk of bias. Additionally, less than one-fifth (17.9%) of the included studies had a high risk of bias.

### Prevalence of unintended pregnancy in Ethiopia

In this study, the overall prevalence of unintended pregnancy in Ethiopia was 28% (95% CI: 25–31) (Fig 2). We had conducted subgroup analyses to investigate how the prevalence of unintended pregnancy varies across different subgroups of studies (Table 2). Consequently, the pooled prevalence of unintended pregnancy was 26.4% (20.8–32.4) and 30.0% (26.6–33.6) for community-based cross-sectional and institution-based cross-sectional studies, respectively. The subgroup analysis based on the region where studies were conducted also showed that the highest prevalence of unintended pregnancy was observed from the Oromiya region [33.8% (29.0–38.7)] followed by SNNPR [30.6% (25.2–36.2)] and Addis Ababa [28.0% (14.0–44.7)]. The lowest prevalence was also noted in Harar [22.9% (20.3–25.5)]. Furthermore, the pooled prevalence of unintended pregnancy was 28.2% (25.1–31.4) and 28.8% (23.7–34.0) for studies conducted prior and post 2014, respectively.

We assessed the issue of publication bias by visual inspection of funnel plot and by using the Egger's regression test. Though the funnel plot looks asymmetrical (Fig 3), the Egger's test showed that no relationship between the effect size and its precision (P-value = 0.776). This might be due to heterogeneity in true effects.

### Determinants of unintended pregnancy in Ethiopia

The significant determinants of unintended pregnancy reported from each study are presented in (S2 Table).

In six studies, researchers examined the association between spousal communication about family planning and unintended pregnancy. Of this, five studies [25,36,37,39,42] showed that spousal communication about family planning was associated with an unintended pregnancy. The result of this meta-analysis showed that not communicating with one's husband about family planning was more likely to lead to unintended pregnancy (OR: 3.56, 95%CI: 1.68–7.53) (Fig 4). Ten studies were examined the association between the use of modern family planning methods and unintended pregnancy[10,21,25,30,34,37,38,40,42,43]. Consequently, the result of this study showed that unintended pregnancy is more likely among women who never used family planning methods (OR: 2.08, 95%CI: 1.18–3.69) (Fig 5).

Of 21 studies that examined the association of maternal education level with unintended pregnancy, ten studies reported that maternal education status was associated with an unintended pregnancy [30–32,34–36,38,40,41]. Furthermore, among the included studies, women's age, time to reach the nearest health facility, household wealth, and marital status showed significant association with unintended pregnancy.

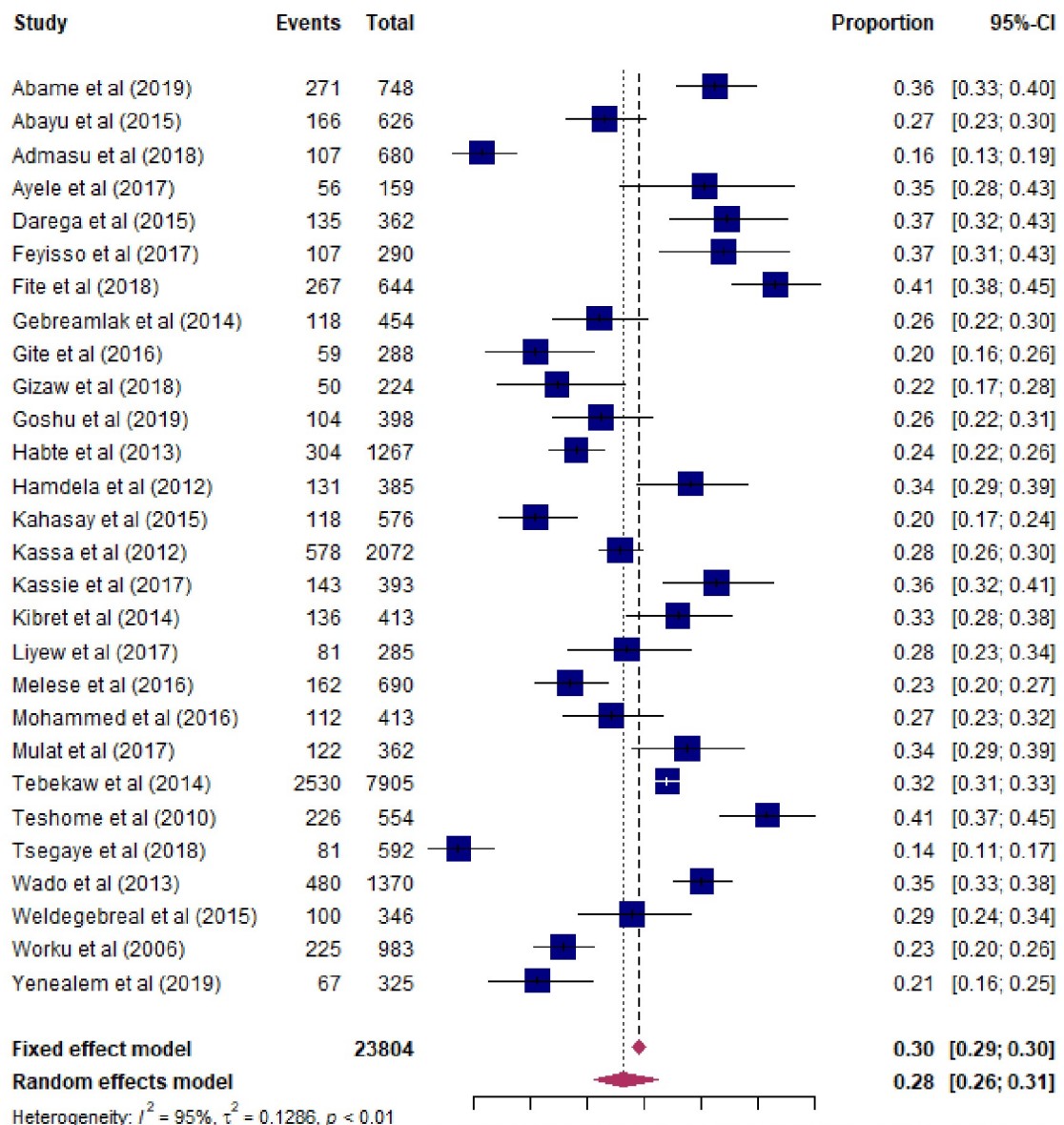

**Fig 2. Forest plot of the pooled prevalence of unintended pregnancy in Ethiopia.**

## Discussion

Unintended pregnancy has significant consequences for the health and welfare of women and children [7]. Pregnancy intention in developing countries is influenced by socio-cultural, environmental, individual and health service-related factors [44]. This systematic review and meta-analysis was conducted to determine the overall prevalence of unintended pregnancy and to identify its determinants in Ethiopia.

We found that the overall prevalence of unintended pregnancy in Ethiopia was 28% (95% CI: 26–31). This finding is consistent with the 2016 EDHS result [4], which reported that from all births in the past 5 years and current pregnancies, one-fourth (25%) were unintended. Additionally, in the previous two consecutive EDHS reports, we noted that the proportion of women who want no more births declined from 42% in 2005 to 37% in 2011. The prevalence

**Table 2. Subgroup analysis of studies included in meta-analysis on prevalence and determinants of unintended pregnancy in Ethiopia.**

| Subgroup | Random effects (95%CI) | Test of heterogeneity ($I^2$) |
|---|---|---|
| **By study design** | | |
| CBCS | 26.4% (20.8–32.4) | 96.6% |
| IBCS | 30.0% (26.6–33.6) | 84.4% |
| Overall | 28.2% (24.1–32.5) | 82.5% |
| **By region** | | |
| Amhara | 24.9% (19.2–31.0) | 95.2% |
| Oromiya | 33.8% (29.0–38.7) | 91.2% |
| SNNPR | 30.6% (25.2–36.2) | 87.6% |
| Addis Ababa | 28.0% (14.0–44.7) | 96.6% |
| National level | 28.0% (20.5–36.1) | 97.1% |
| Tigray | 26.5% (23.1–30.0) | - |
| Harar | 22.9% (20.3–25.5) | - |
| Overall | 26.5%(24.8–28.2) | 88.8% |
| **By data collection period** | | |
| 2014 and before | 28.2% (25.1–31.4) | 94.6% |
| After 2014 | 28.8% (23.7–34.0) | 94.5% |
| Overall | 28.4% (25.7–31.1) | 87.2% |

of unintended pregnancy found in this study agreed with other previous studies conducted in Egypt [45], Ghana [46], and Bangladesh [9], which reported that nearly one-third (30.7%), 29.8% and 29% of pregnancies were unintended, respectively. However, the magnitude of unintended pregnancy found in this study was lower than previous studies conducted in Malawi, South Africa, and the Republic of Congo [47–50]. This variation might be attributed to methodological differences in the assessment of pregnancy intention. The other possible

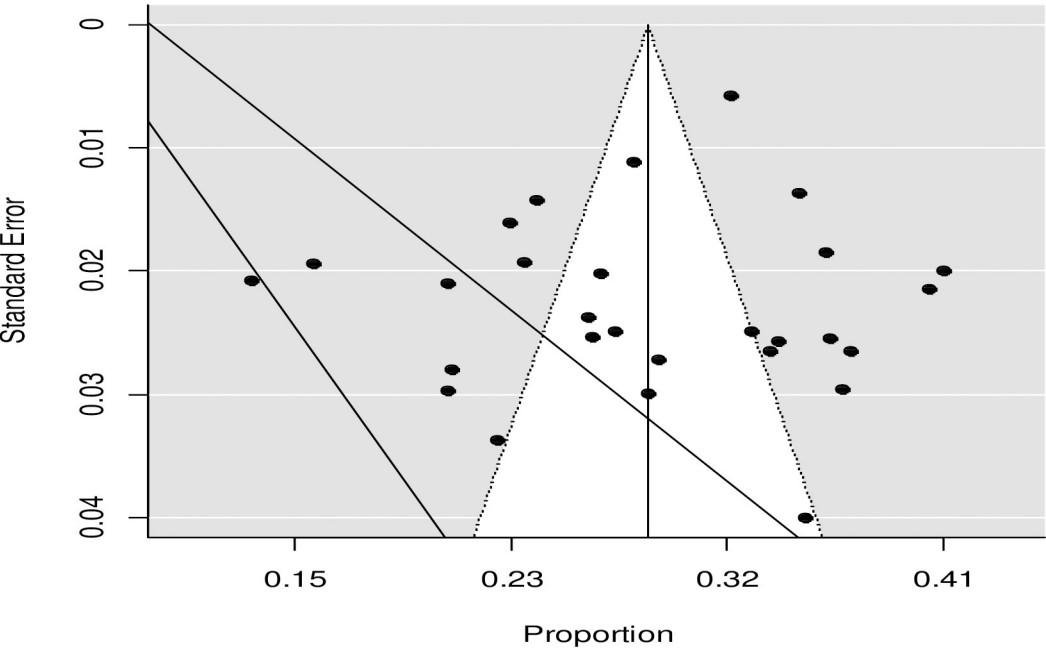

**Fig 3. Funnel plot of the prevalence of unintended pregnancy in Ethiopia.**

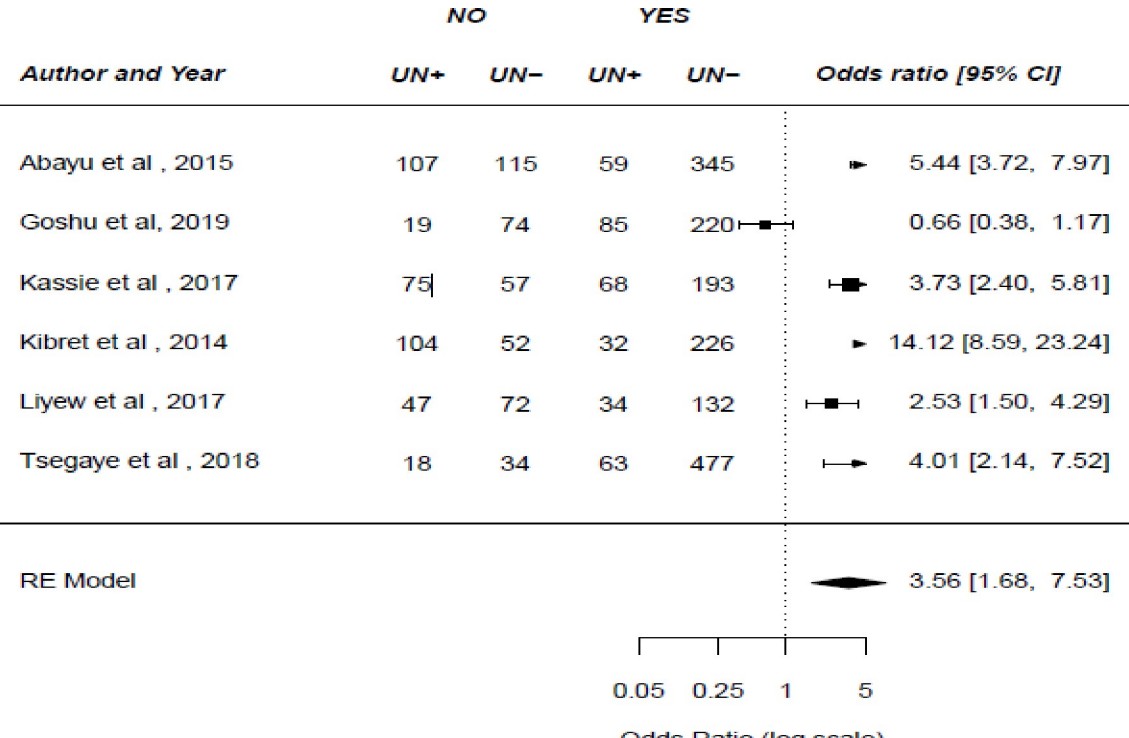

**Fig 4. The pooled odds ratio of the association between spousal communication about family planning and unintended pregnancy in Ethiopia.**

reason for inconsistency in the prevalence of unintended pregnancy could be explained by the difference in the socio-economic characteristics of study participants [51–53].

In this study, we found that the pooled prevalence of unintended pregnancy was 26.6% and 30% for community-based cross-sectional and institutional-based cross-sectional studies, respectively. This variation could be explained by institution-based studies that include women who visit health facilities for legal abortion services, which is probably unintended pregnancy. About 97% of women seeking an abortion reported having an unintended pregnancy [51]. The highest prevalence of unintended pregnancy was observed from the Oromiya region (33.8%), and the lowest was in Harar (22.9%). The possible explanation might be due to the difference in the utilization of family planning methods.

In this systematic review and meta-analysis, we noted that a number of factors were found to be associated with an unintended pregnancy. Consequently, we found that women who hadn't communicated with their husbands about family planning methods were more likely to have an unintended pregnancy. The possible explanation for this result might be women's perception that their husbands oppose family planning, which is one of the dominant factors for discouraging the contraceptive practice in a wide variety of settings [19]. Partner awareness of contraceptives and open discussion about family planning methods will decrease the risk of unintended pregnancy. The result of this study also showed that unintended pregnancy is more likely among women who never used family planning methods. This is due to the fact that awareness and proper utilization of modern contraceptives are crucial to reduce unintended pregnancy.

In this systematic review, unintended pregnancy was less likely among married women. The possible reason for this result might be in Ethiopian culture, pregnancy without marriage

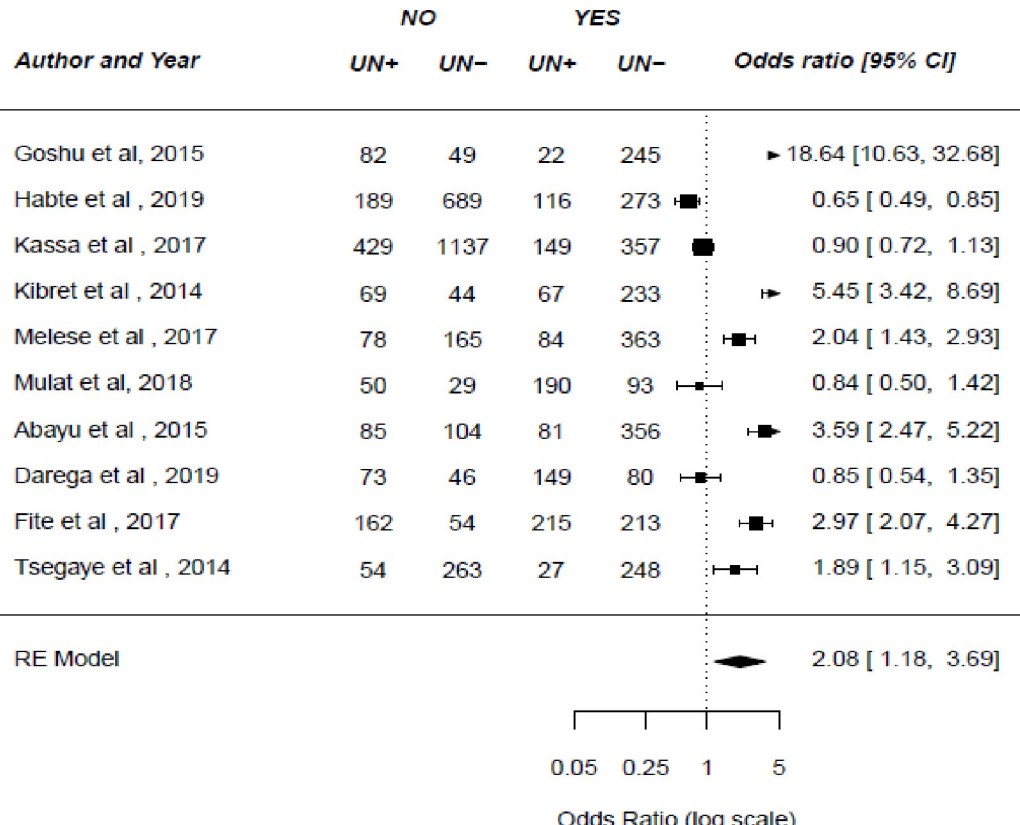

**Fig 5. The pooled odds ratio of the association between the use of family planning and unintended pregnancy in Ethiopia.**

is unacceptable in most communities. This result is supported by previous studies conducted in South Africa and Kenya [54,55].

Though previous evidence stated that youngest women experience the highest rate of unintended pregnancy [56], in this systematic review, we observed controversial results regarding the association between women's age and unintended pregnancy. Some of the studies reported that younger women are more likely to have an unintended pregnancy, while other studies showed that older women are more likely to have unintended pregnancy.

Due to the inconsistent classification of maternal education level, time-lapse to reach the nearest health facility, parity, occupation and religion, we couldn't pool quantitatively, in this systematic review and meta-analysis. Contradictory results are observed between individual studies. Four studies [30,32,36,41] reported that women who have relatively better education were less likely to have an unintended pregnancy as compared to those who didn't have formal education. It is reasonable that as educational level increases, the awareness of reproductive health also increases. Additionally, as the educational level increases, the spousal communication on pregnancy, awareness of long-term family planning, and receiving adequate ANC also increases. However, two studies reported that women who had relatively better education were more likely to have unintended pregnancy [31,40].

Moreover, in this study, we observed that as the time-lapse to reach the nearest health facility for providing contraceptives increased, women were more likely to have unintended pregnancy [10,30,43]. It is also reasonable that as the time-lapse to reach the nearest health facility

increases, the probability of missing ANC visits also increase, which in turn is associated with unintended pregnancy [13,19].

In this study, we noted that the parity of respondents showed a significant relationship with an unintended pregnancy. Consequently, multiparous women were more likely to experience an unintended pregnancy than nulliparous [34,35,41,57]. This might be because of women who have attained their desired number of children will perceive any additional child as unwanted [58]. Furthermore, the pregnancy of unemployed women is more likely to be unintended than employed women [30,37].

Moreover, in this study, another important variable which significantly associated with unintended pregnancy was religion. Muslim mothers were less likely to report having an unintended pregnancy as compared to orthodox mothers. This finding is consistent with a study conducted in Ghana [59]. This might be as a result of doctrinal differences among the women, along with different religions.

## Limitations

This study was not without limitations. Firstly, the review was limited to only articles published in the English language. Secondly, all of the included studies were cross-sectional, which limits assessment of the cause-effect relationships. Thirdly, we were unable to show the pooled odds ratio for all variables associated with unintended pregnancy because the included studies classified the variables in different ways.

## Conclusions

In this study, a high prevalence of unintended pregnancy was observed. Lack of spousal communication, never using family planning, maternal education, and household wealth level were significantly associated with an unintended pregnancy. This study implies the need to develop plans and policies to improve the awareness of contraceptive utilization and strengthen spousal communication related to pregnancy. Emphasis should be given to those women at a distant from health facilities, unmarried and teenagers.

## Supporting information

**S1 Checklist.**
(DOC)

**S1 File. List of excluded references and reasons for exclusion.**
(DOCX)

**S1 Table. Assessing the risk of bias for the included studies.**
(XLSX)

**S2 Table. Significant determinants of unintended pregnancy reported from each study.**
(DOCX)

## Author Contributions

**Conceptualization:** Muluneh Alene, Leltework Yismaw, Bekalu Kassie, Reta Yeshambel.

**Data curation:** Muluneh Alene, Leltework Yismaw, Yebelay Berelie, Bekalu Kassie, Moges Agazhe Assemie.

**Formal analysis:** Muluneh Alene, Leltework Yismaw, Yebelay Berelie, Moges Agazhe Assemie.

**Investigation:** Muluneh Alene, Bekalu Kassie, Reta Yeshambel.

**Methodology:** Muluneh Alene, Leltework Yismaw, Yebelay Berelie, Reta Yeshambel, Moges Agazhe Assemie.

**Resources:** Muluneh Alene, Reta Yeshambel, Moges Agazhe Assemie.

**Software:** Muluneh Alene, Leltework Yismaw, Yebelay Berelie, Moges Agazhe Assemie.

**Supervision:** Yebelay Berelie, Reta Yeshambel, Moges Agazhe Assemie.

**Validation:** Muluneh Alene, Leltework Yismaw, Reta Yeshambel.

**Visualization:** Muluneh Alene, Leltework Yismaw, Yebelay Berelie, Bekalu Kassie, Reta Yeshambel, Moges Agazhe Assemie.

**Writing – original draft:** Muluneh Alene, Leltework Yismaw, Bekalu Kassie.

**Writing – review & editing:** Muluneh Alene, Leltework Yismaw, Yebelay Berelie, Bekalu Kassie, Reta Yeshambel, Moges Agazhe Assemie.

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
