## [Decision Letter · Decision Letter 0]

1 Oct 2019

PONE-D-19-21662

Prevalence and determinants of unintended pregnancy in Ethiopia: a systematic review and meta-analysis of observational studies

PLOS ONE

Dear Mr. Addis,

Thank you for submitting your manuscript to PLOS ONE. After careful consideration, we feel that it has merit but does not fully meet PLOS ONE’s publication criteria as it currently stands. Therefore, we invite you to submit a revised version of the manuscript that addresses the points raised during the review process.

We would appreciate receiving your revised manuscript by Nov 15 2019 11:59PM. To enhance the reproducibility of your results, we recommend that if applicable you deposit your laboratory protocols in protocols.io, where a protocol can be assigned its own identifier (DOI) such that it can be cited independently in the future. For instructions see: http://journals.plos.org/plosone/s/submission-guidelines#loc-laboratory-protocols

We look forward to receiving your revised manuscript.

Kind regards,

Seth Adu-Afarwuah

Academic Editor

PLOS ONE

Journal Requirements:

2. Please confirm that you have included all items recommended in the PRISMA checklist including details of reasons for study exclusions in the PRISMA flowchart and number of studies excluded for each reason, the dates of the search, and the full electronic search strategy used to identify studies with all search terms and limits for at least one database.

Additional Editor Comments (if provided):

Reviewers' comments:

Reviewer's Responses to Questions

**Comments to the Author**

1. Is the manuscript technically sound, and do the data support the conclusions?

Reviewer #1: Yes

Reviewer #2: Yes

Reviewer #3: Partly

Reviewer #4: Yes

Reviewer #5: Yes

2. Has the statistical analysis been performed appropriately and rigorously? 

Reviewer #1: I Don't Know

Reviewer #2: Yes

Reviewer #3: Yes

Reviewer #4: Yes

Reviewer #5: Yes

3. Have the authors made all data underlying the findings in their manuscript fully available?

Reviewer #1: Yes

Reviewer #2: Yes

Reviewer #3: Yes

Reviewer #4: No

Reviewer #5: Yes

4. Is the manuscript presented in an intelligible fashion and written in standard English?

Reviewer #1: Yes

Reviewer #2: No

Reviewer #3: Yes

Reviewer #4: No

Reviewer #5: Yes

5. Review Comments to the Author

Reviewer #1: I do not have major comments on this article. The objective and findings are generally clear. The authors identified the prevalence of unintended pregnancy in Ethiopia, and associated risk factors.

Minor suggestion : delete first sentence of abstract - many people may take offense to the claim that pregnancies should be planned. Unintended pregnancy is well recognised, but this term is not interchangeable with the claim that all pregnancies should be planned. Women are allowed to choose whether they want to plan their pregnancies. Delete the dfirst sentence, and this should fix the problem.

The quality of the English needs major work.

Reviewer #2: 1. Was a time period determined for the search? If yes, what was it? If no, then please mention that.

2. How did you search for unpublished studies?

3. Was any attempt made to contact authors of studies for which full papers were not obtained?

4. The writing needs quite a bit of improvement in terms of flow. It is not an easy and smooth read. There is a lot of repetition. Sentences need to be clearer and more precise. The use of the word "Accordingly" in several places does not make sense. Also, punctuation needs to be revised.

Reviewer #3: This well written manuscript is a systematic review and meta-analysis of observational studies prevalence and determinants of unintended pregnancy in Ethiopia. This review addresses an important Issue (prevalence of unintended pregnancies) for Ethiopia but may not be highly relevant to the general readers of the journal.

The methodology sounds fine; however, I have a few concerns:

1. Literature search: it is limited to 2 data bases and google scholar, which is not comprehensive. The authors should have searched more relevant data bases. They mentioned that they searched national data bases, however, there is no description of such data base. The search was limited to English language. As this study is investigating the prevalence of unintended pregnancy in a specific county/ region, it must have been included the articles in local language. This may increase the risk of reporting and publication bias.

2. Statistical analysis: authors should explain the reasons in heterogeneity among studies. Why it exists, and how it affects the credibility of the overall results?

3. Discussion: there is no mention of women’s age, time to reach the nearest health facility, household wealth, being married… in results section, but they show up in dissuasion as the potential determinants on the prevalence of unintended pregnancies

I have not checked any original studies for double checking of the abstracted data, nor perform any literature review to rechecking included studies.

Reviewer #4: This is an interesting review on prevalence and determinants of unintended pregnancy in Ethiopia. Below I provide some comments which may help in improving the manuscript:

Please provide reference for unintended pregnancy definition.

Parity, occupation, religion and economic status are important determinants of unintended pregnancy. Could the authors provide insight into why they were not significantly associated with unintended pregnancy in the review?

As readers may not be familiar with Ethiopia, please define SNNPR at first instance.

Reviewer #5: There are 195 English language corrections that I strongly suggest you follow in the Comments section of the attached PDF file. This is an important topic, and the research is very valuable for the public health of Ethiopia.

6. PLOS authors have the option to publish the peer review history of their article (what does this mean?). If published, this will include your full peer review and any attached files.

Reviewer #1: No

Reviewer #2: No

Reviewer #3: No

Reviewer #4: No

Reviewer #5: No

---

## [Author Response · Author response to Decision Letter 0]

19 Oct 2019

Author's response to reviews

Title: Prevalence and determinants of unintended pregnancy in Ethiopia: a systematic review and meta-analysis of observational studies

Authors 

Author’s email addresses 

MA: mulunehadis@gmail.com

LY: lielt.yismaw@gmail.com

YB: yebelay.ma@gmail.com

BK: bekalukassiedmu@gmail.com

RY: reta.yeshambel@gmail.com

MAA: agazhemoges@gmail.com

Date: 17 October 2019

Dear Editor,

We thank you for the chance to resubmit our revised manuscript. Also, we would like to thank the reviewers for sharing the view and experience. The comments are very important that will improve the manuscript. The point-by-point responses for each of the comments are provided in the following pages. We hope that the revisions meet your standards and that the paper would be published in your journal. We look forward to working with you towards a final published product.

Sincerely,

Muluneh Alene, MPH

On behalf of co-authors

Point by point responses to queries 

Reviewer#1

Comments and points raised Authors response

1. “Delete first sentence of abstract?” Answer: thank you dear reviewer for your constructive comments!

o Based on the suggestion given, we make modification on the first sentence of the abstract section in the revised form of the manuscript.

2. “The quality of the English needs major work.”

 Answer: Thank you!

o The issue of English language is resolved using English language professionals in the revised form of the manuscript.

Reviewer#2

Comments and points raised Authors response

1. “Was a time period determined for the search? If yes, what was it? If no, then please mention that.”

 Answer: thank you dear reviewer for your constructive comments!

o The searching time period was already stated and it was between the 15th of February and 30th of April 2019

2. “How did you search for unpublished studies?”

 Answer: Thank you!

o To find unpublished papers relevant to this study, some research centers including Addis Ababa digital library were searched.

3. “Was any attempt made to contact authors of studies for which full papers were not obtained?”

 Answer: Thank you!

o We were planned to contact the primary author by email at least three times if a study is not fully available. But, studies searched for this systematic review and meta-analysis were fully obtained. 

4. “The writing needs quite a bit of improvement in terms of flow.” Answer: Thank you!

o Based on your suggestion, we revised the flow of idea in the revised form of the manuscript and we make highlight for changes.

Reviewer#3

Comments and points raised Authors response

1. “Literature search: it is limited to 2 data bases and Google scholar, which is not comprehensive. The authors should have searched more relevant databases. They mentioned that they searched national databases, however, there is no description of such database. The search was limited to English language. As this study is investigating the prevalence of unintended pregnancy in a specific county/ region, it must have been included the articles in local language. This may increase the risk of reporting and publication bias.”

 Answer: thank you dear reviewer for your constructive comments!

o Based on the comments given and to make comprehensive, in the revised form of the manuscript, we extend our searches for studies from PubMed/MEDLINE, Web of Science, CINAHL, Google Scholar, Science Direct and Cochrane Library.

o We were searched the “Ethiopian Journal of Public Health and Nutrition” for the national databases.

o You are right dear reviewer; not including articles published in local language may increase the risk of reporting and publication bias.

o But, we were searched for articles published in local language related to this study, but we can’t get such articles.

2. “Authors should explain the reasons in heterogeneity among studies. Why it exists, and how it affects the credibility of the overall results?”

 Answer: Thank you!

o Based on the comment given, we included the possible reasons for heterogeneity among included studies.

o In this systematic review and meta-analysis, the possible sources of heterogeneity among studies are differences in study participants, study design, risk of bias and data collection period.

o To account the heterogeneity among the included studies, subgroup analysis and a random effect meta-analysis with an estimation of DerSimonian and Laird method was performed.

3. “Discussion: there is no mention of women’s age, time to reach the nearest health facility, household wealth, being married… in results section, but they show up in dissuasion as the potential determinants on the prevalence of unintended pregnancies”

 o Based on the comments given, we included the stated variables in the result section.

Reviewer#4

Comments and points raised Authors response

1. “Parity, occupation, religion and economic status are important determinants of unintended pregnancy. Could the authors provide insight into why they were not significantly associated with unintended pregnancy in the review?” Answer: thank you dear reviewer for your constructive comments!

o In the revised form of the manuscript, we already considered the association between the mentioned variables and unintended pregenancy.

2. “As readers may not be familiar with Ethiopia, please define SNNPR at first instance.”

 Answer: Thank you!

o Based on the comment given, we already define the word “SNNPR” at first instance in the revised form of the manuscript.

Reviewer#5

Comments and points raised Authors response

1. “There are 195 English language corrections that I strongly suggest you follow in the Comments section of the attached PDF file.” Answer: thank you dear reviewer for your constructive comments!

o We thank you for your revision, and we considered all English language corrections in the revised form of the manuscript. 

Thank you!!!

---

## [Decision Letter · Decision Letter 1]

6 Nov 2019

PONE-D-19-21662R1

Prevalence and determinants of unintended pregnancy in Ethiopia: a systematic review and meta-analysis of observational studies

PLOS ONE

Dear Mr. Addis,

Thank you for submitting your manuscript to PLOS ONE. After careful consideration, we feel that it has merit but does not fully meet PLOS ONE’s publication criteria as it currently stands. Therefore, we invite you to submit a revised version of the manuscript that addresses the points raised during the review process.

We would appreciate receiving your revised manuscript by Dec 21 2019 11:59PM. To enhance the reproducibility of your results, we recommend that if applicable you deposit your laboratory protocols in protocols.io, where a protocol can be assigned its own identifier (DOI) such that it can be cited independently in the future. For instructions see: http://journals.plos.org/plosone/s/submission-guidelines#loc-laboratory-protocols

We look forward to receiving your revised manuscript.

Kind regards,

Seth Adu-Afarwuah

Academic Editor

PLOS ONE

Reviewers' comments:

Reviewer's Responses to Questions

**Comments to the Author**

1. If the authors have adequately addressed your comments raised in a previous round of review and you feel that this manuscript is now acceptable for publication, you may indicate that here to bypass the “Comments to the Author” section, enter your conflict of interest statement in the “Confidential to Editor” section, and submit your "Accept" recommendation.

Reviewer #1: All comments have been addressed

Reviewer #2: (No Response)

Reviewer #4: All comments have been addressed

Reviewer #5: (No Response)

2. Is the manuscript technically sound, and do the data support the conclusions?

Reviewer #1: Yes

Reviewer #2: Yes

Reviewer #4: Yes

Reviewer #5: Yes

3. Has the statistical analysis been performed appropriately and rigorously? 

Reviewer #1: I Don't Know

Reviewer #2: Yes

Reviewer #4: Yes

Reviewer #5: Yes

4. Have the authors made all data underlying the findings in their manuscript fully available?

Reviewer #1: Yes

Reviewer #2: Yes

Reviewer #4: Yes

Reviewer #5: Yes

5. Is the manuscript presented in an intelligible fashion and written in standard English?

Reviewer #1: Yes

Reviewer #2: Yes

Reviewer #4: Yes

Reviewer #5: No

6. Review Comments to the Author

Reviewer #1: No further comments.

xxxxxxxxxxxxxxxxxxxxxxxxxxxxxxxxxxxxxxxxxxxxxxxxxxxxxxxxxxxxxxxxxxxxxxxxxxxxxxx

Reviewer #2: The authors have not addressed the comments I have raised in my previous review.

The time period of the search is still not mentioned. Also in the flowchart, the further breakdown of how many studies were excluded because the full study was not available to them and how many did not have the outcome of interest is needed. What all efforts were done to try and get the full study? Were the authors contacted?

Reviewer #4: (No Response)

Reviewer #5: The authors have done a wonderful job making so many suggested English grammar corrections. The manuscript is significantly improved. However, there are still 12 more minor corrections remaining. See the comments in the attached file.

7. PLOS authors have the option to publish the peer review history of their article (what does this mean?). If published, this will include your full peer review and any attached files.

Reviewer #1: No

Reviewer #2: No

Reviewer #4: No

Reviewer #5: No

---

## [Author Response · Author response to Decision Letter 1]

27 Nov 2019

Author's response to reviews

Title: Prevalence and determinants of unintended pregnancy in Ethiopia: a systematic review and meta-analysis of observational studies

Authors 

Author’s email addresses 

MA: mulunehadis@gmail.com

LY: lielt.yismaw@gmail.com

YB: yebelay.ma@gmail.com

BK: bekalukassiedmu@gmail.com

RY: reta.yeshambel@gmail.com

MAA: agazhemoges@gmail.com

Date: 27 November 2019

Dear Editor,

We thank you for the chance to resubmit our revised manuscript. Also, we would like to thank the reviewers for sharing the view and experience. The comments are very important that will improve the manuscript. The point-by-point responses for each of the comments are provided in the following pages. We hope that the revisions meet your standards and that the paper would be published in your journal. We look forward to working with you towards a final published product.

Sincerely,

Muluneh Alene, MPH

On behalf of co-authors

Point by point responses to queries 

Reviewer#2

Comments and points raised Authors response

The time period of the search is still not mentioned. Also in the flowchart, the further breakdown of how many studies were excluded because the full study was not available to them and how many did not have the outcome of interest is needed. What all efforts were done to try and get the full study? Were the authors contacted? Answer: thank you dear reviewer for your constructive comments!

o The searching time period was already stated in the previous revised form of the manuscript: Page 6, line 168-169.

o In Fig. 1 (flow chart diagram), we included the number of studies excluded because the outcome of interest did not report, having data that were not extractable and conducted in other countries.

o We were planned to contact the primary author by email at least three times if a study is not fully available. But, studies searched for this systematic review and meta-analysis were fully obtained.

Reviewer#5

Comments and points raised Authors response

1. “There are still 12 more minor corrections remaining.”

 Answer: thank you dear reviewer for your constructive comments!

o We thank you again for your revision, and we considered all English language corrections in the revised form of the manuscript. 

o In addition, we used English language professionals to write the manuscript in Standard English.

---

## [Decision Letter · Decision Letter 2]

10 Jan 2020

PONE-D-19-21662R2

Prevalence and determinants of unintended pregnancy in Ethiopia: a systematic review and meta-analysis of observational studies

PLOS ONE

Dear Mr. Addis,

Thank you for submitting your manuscript to PLOS ONE. After careful consideration, we feel that it has merit but does not fully meet PLOS ONE’s publication criteria as it currently stands. Therefore, we invite you to submit a revised version of the manuscript that addresses the points raised during the review process.

We would appreciate receiving your revised manuscript by Feb 24 2020 11:59PM. To enhance the reproducibility of your results, we recommend that if applicable you deposit your laboratory protocols in protocols.io, where a protocol can be assigned its own identifier (DOI) such that it can be cited independently in the future. For instructions see: http://journals.plos.org/plosone/s/submission-guidelines#loc-laboratory-protocols

We look forward to receiving your revised manuscript.

Kind regards,

Seth Adu-Afarwuah

Academic Editor

PLOS ONE

Reviewers' comments:

Reviewer's Responses to Questions

**Comments to the Author**

1. If the authors have adequately addressed your comments raised in a previous round of review and you feel that this manuscript is now acceptable for publication, you may indicate that here to bypass the “Comments to the Author” section, enter your conflict of interest statement in the “Confidential to Editor” section, and submit your "Accept" recommendation.

Reviewer #2: (No Response)

2. Is the manuscript technically sound, and do the data support the conclusions?

Reviewer #2: Yes

3. Has the statistical analysis been performed appropriately and rigorously? 

Reviewer #2: I Don't Know

4. Have the authors made all data underlying the findings in their manuscript fully available?

Reviewer #2: Yes

5. Is the manuscript presented in an intelligible fashion and written in standard English?

Reviewer #2: Yes

6. Review Comments to the Author

Reviewer #2: Line 73 "...25% WERE unintended..."

Line 82 it is not 100% that a woman WILL have low mental and physical health, I would reword it and say that women with unintended pregnancy are more likely to have low mental and physical health.

My question about the time period of articles that were searched still remains unclear, the time period mentioned 15 February to 30 April 2019 is the time period during which the search was conducted or only articles published within this time period was included? Because if it is the latter, which I doubt it is, then that is a very very short time period.

This needs to be clearly mentioned in the article.

7. PLOS authors have the option to publish the peer review history of their article (what does this mean?). If published, this will include your full peer review and any attached files.

Reviewer #2: Yes: Tarannum Behlim

---

## [Author Response · Author response to Decision Letter 2]

28 Jan 2020

Author's response to reviews

Title: Prevalence and determinants of unintended pregnancy in Ethiopia: a systematic review and meta-analysis of observational studies

Authors 

Author’s email addresses 

MA: mulunehadis@gmail.com

LY: lielt.yismaw@gmail.com

YB: yebelay.ma@gmail.com

BK: bekalukassiedmu@gmail.com

RY: reta.yeshambel@gmail.com

MAA: agazhemoges@gmail.com

Date: 25 January 2020

Dear Editor,

We thank you for the chance to resubmit our revised manuscript. Also, we would like to thank the reviewers for sharing the view and experience. The comments are very important that will improve the manuscript. The point-by-point responses for each of the comments are provided in the following pages. We hope that the revisions meet your standards and that the paper would be published in your journal. We look forward to working with you towards a final published product.

Sincerely,

Muluneh Alene, MPH

On behalf of co-authors

Point by point responses to queries 

Reviewer#2

Comments and points raised Authors response

1. Line73"...25% were unintended...", Line 82 it is not 100% that a woman WILL have low mental and physical health, I would reword it and say that women with unintended pregnancy are more likely to have low mental and physical health

 Answer: thank you dear reviewer for your constructive comments!

In the revised form of the manuscript, we already incorporate your suggestion. 

2. My question about the time period of articles that were searched still remains unclear, the time period mentioned 15 February to 30 April 2019 is the time period during which the search was conducted or only articles published within this time period was included? Because if it is the latter, which I doubt it is, then that is a very short time period. This needs to be clearly mentioned in the article. Answer: thank you dear reviewer for your constructive comments which are very important to improve the manuscript!

The time period what we mentioned in page 6, Line 169-170 (15 February to 30 April 2019) is only the period what we search for studies.

In this study, we included all existing articles without considering the year of publication. So, as it indicates in page 6, Line 177-178 the publication year among the included studies was between 2006 and 2019.

We make highlight for changes in the revised form of the manuscript. 

Thank you!!!

---

## [Decision Letter · Decision Letter 3]

16 Mar 2020

Prevalence and determinants of unintended pregnancy in Ethiopia: a systematic review and meta-analysis of observational studies

PONE-D-19-21662R3

Dear Dr. Addis,

We are pleased to inform you that your manuscript has been judged scientifically suitable for publication and will be formally accepted for publication once it complies with all outstanding technical requirements.

With kind regards,

Seth Adu-Afarwuah

Academic Editor

PLOS ONE

Additional Editor Comments (optional):

Reviewers' comments:

Reviewer's Responses to Questions

**Comments to the Author**

1. If the authors have adequately addressed your comments raised in a previous round of review and you feel that this manuscript is now acceptable for publication, you may indicate that here to bypass the “Comments to the Author” section, enter your conflict of interest statement in the “Confidential to Editor” section, and submit your "Accept" recommendation.

Reviewer #2: All comments have been addressed

2. Is the manuscript technically sound, and do the data support the conclusions?

Reviewer #2: Yes

3. Has the statistical analysis been performed appropriately and rigorously? 

Reviewer #2: I Don't Know

4. Have the authors made all data underlying the findings in their manuscript fully available?

Reviewer #2: Yes

5. Is the manuscript presented in an intelligible fashion and written in standard English?

Reviewer #2: Yes

6. Review Comments to the Author

Reviewer #2: (No Response)

7. PLOS authors have the option to publish the peer review history of their article (what does this mean?). If published, this will include your full peer review and any attached files.

Reviewer #2: Yes: Tarannum Behlim